# Metformin Mitigates Nickel-Elicited Angiopoietin-Like Protein 4 Expression via HIF-1α for Lung Tumorigenesis

**DOI:** 10.3390/ijms21020619

**Published:** 2020-01-17

**Authors:** Yu-Ting Kang, Wen-Cheng Hsu, Chu-Chyn Ou, Hui-Chun Tai, Hui-Ting Hsu, Kun-Tu Yeh, Jiunn-Liang Ko

**Affiliations:** 1Institute of Medicine, Chung-Shan Medical University, Taichung 402, Taiwan; polove908@gmail.com; 2Department of Endocrinology and Metabolism, Tungs’ Taichung Metro Harbor Hospital, Taichung 435, Taiwan; t3526@ms.sltung.com.tw; 3School of Nutrition, Chung Shan Medical University, Taichung 402, Taiwan; occ@csmu.edu.tw; 4Department of Pathology, Changhua Christian Hospital, Changhua 500, Taiwan; 121832@cch.org.tw (H.-C.T.); 128202@cch.org.tw (H.-T.H.); 5Department of Medical Oncology and Chest Medicine, Chung Shan Medical University Hospital, Taichung 402, Taiwan

**Keywords:** Nickel, angiopoietin-like protein 4, hypoxia-inducible factor 1 alpha, metformin, chemoprevention

## Abstract

Nickel (Ni), which is a carcinogenic workplace hazard, increases the risk of lung cancer. Angiopoietin-like protein 4 (ANGPTL4) is a multifunctional cytokine that is involved in both angiogenesis and metastasis, but its role in lung cancer is still not clear. In this study, we assessed the role of ANGPTL4 in lung carcinogenesis under nickel exposure and investigated the effects of the antidiabetic drug metformin on ANGPTL4 expression and lung cancer chemoprevention. Our results showed that ANGPTL4 is increased in NiCl_2_-treated lung cells in a dose- and time-course manner. The expression of ANGPTL4 and HIF-1α induced by NiCl_2_ were significantly repressed after metformin treatment. The downregulation of HIF-1α expression by ROS savenger and HIF-1α inhibitor or knockdown by lentiviral shRNA infection diminished NiCl_2_-activated ANGPTL4 expression. Chromatin immunoprecipitation and the luciferase assay revealed that NiCl_2_-induced HIF-1α hypoxia response element interactions activate ANGPTL4 expression, which is then inhibited by metformin. In conclusion, the increased presence of ANGPTL4 due to HIF-1α accumulation that is caused by nickel in lung cells may be one mechanism by which nickel exposure contributes to lung cancer progression. Additionally, metformin has the ability to prevent NiCl_2_-induced ANGPTL4 through inhibiting HIF-1α expression and its binding activity. These results provide evidence that metformin in oncology therapeutics could be a beneficial chemopreventive agent.

## 1. Introduction

Nickel (Ni) compounds are common environmental pollutants, the sources of which include metallurgy waste air, burning of fossil fuels, and cigarette smoke, and they are recognized as a group 1 human carcinogen by the International Agency for Research on Cancer [1]. Humans are usually exposed to nickel through inhalation, skin absorption, and oral consumption, with the kidneys, lungs, and liver being the major organs for nickel accumulation [2,3]. Previous in vitro and in vivo studies on nickel toxicity and carcinogenesis have demonstrated its ability to induce DNA damage, epigenetic alterations, disruption of cellular iron homeostasis, and generation of reactive oxygen species (ROS), although nickel has been found to be only weakly mutagenic. However, there is a high incidence of lung, nasal, and pharyngeal cancer among refinery workers after long-term exposure to nickel [4]. Animal studies have revealed that the inhalation of nickel increases tumour formation in rats [5]. Nickel acts as an imitator of hypoxia, with the resulting hypoxia responding to activation and hypoxia-inducible factor (HIF) accumulation, which inhibits the activity of prolyl hydroxylases and disrupts the degradation of HIFs [6,7]. The hypoxic condition is fundamental to carcinogenesis and it promotes angiogenesis, glucose uptake, inflammation, and DNA mutation. Our previous studies have indicated that nickel exposure results in ROS production, which consequently contributes to epithelial–mesenchymal transition (EMT) and the activation of autophagy [8,9]. In this study, we propose a therapeutic strategy against HIF-1-induced carcinogenesis due to nickel exposure.

Angiopoietin-like protein 4 (ANGPTL4) exhibits structural similarity to the multifunctional angiopoietins, which are involved in glucose homeostasis, lipid metabolism, angiogenesis, inflammation, and tumour progression and metastasis [10]. It has been demonstrated that ANGPTL4 is upregulated by fasting conditions, hypoxia, transforming growth factor-β (TGF-β), and prostaglandin-E2, and it is transcriptionally regulated by the peroxisome proliferator-activated receptors β and δ [11,12,13,14]. ANGPTL4 serves an important role in the tumour microenvironment, especially that which is hypoxia-induced. Hypoxia promotes uveal melanoma cell angiogenesis and metastasis via the upregulation of ANGPTL4 and VEGF expression [15]. It was also reported that the blockage of HIF-mediated ANGPTL4 inhibits vascular metastasis of breast cancer cells to the lungs [16]. Further evidence has demonstrated that ANGPTL4 can be a potential diagnostic or prognostic marker for numerous cancers. However, the vital mechanism of ANGPTL4 in nickel-induced carcinogenesis remains unknown.

Metformin is the most widely prescribed agent for patients with hyperglycaemia. The primary functions of antihyperglycaemia agents include enhancing insulin sensitivity, inhibiting hepatic gluconeogenesis, and increasing glucose uptake by peripheral tissues [17]. Further examination of the clinical effects and treatment mechanisms of metformin have been extended to include antitumour, antiaging, and neuroprotective effects, obesity, and an optional treatment for polycystic ovary syndrome [18]. Numerous epidemiological and clinical studies have found that metformin use is associated with a decreased risk of cancer incidence and mortality when compared with other hypoglycaemic drugs. It has been suggested that metformin has potential for use as an adjuvant therapy or chemopreventive agent in the treatment of cancer or in the prevention of carcinogenesis.

In our previous study, metformin suppressed NiCl_2_-induced autophagy and glycolytic enzyme activity that played the roles in lung cancer tumorigenesis [9]. The objective of the present study was to evaluate the role of ANGPTL4 in lung cancer and normal cell lines under NiCl_2_ exposure and investigate the preventive effect of metformin on NiCl_2_-activated ANGPTL4 via ROS accumulation.

## 2. Results

### 2.1. Analysis of ANGPTL4 Expression in Normal and Tumour Lung Cell Lines under NiCl_2_ Exposore

We screened the expression of ANGPTL4 in various normal cell lines BEAS-2B, MRC-5, and WI-38 and tumour lung cell lines A549, H1355, H1299, H1975 CL1-0, and CL1-5 to investigate the roles of ANGPTL4 in lung cancer progression. We observed that most of the cell lines, except CL1-0 and CL1-5, had increased expression of full-length ANGPTL4 (Figure 1A). Additionally, to evaluate the effect of NiCl_2_ on normal and tumour cells, the expression of ANGPTL4 mRNA, and protein was investigated in various normal and tumour lung cell lines that were cultured under the treatment of NiCl_2_ (Figure 1B,C). The expression of ANGPTL4 was significantly upregulated in almost all cell lines under NiCl_2_ exposure.

BEAS-2B cells were treated with varying concentrations of NiCl_2_ and for various periods of time to confirm the effects of nickel treatment on ANGPTL4 expression. As evident in Figure 2A,B, NiCl_2_ exposure resulted in the substantial upregulation of ANGPTL4 protein and gene expression, as assessed while using Western blot and real-time, respectively, in a dose- and time-dependent manner. NiCl_2_ also stimulated HIF-1α expression; we found that HIF-1α showed up after 6 h of NiCl_2_ exposure. Consistently, ANGPTL4 was also expressed after 6 h of exposure and showed obvious expression after 24 h of exposure (Figure 2C,D).

### 2.2. NiCl_2_ Induces Numerous Oncology Genes in Both Malignant and Normal Lung Cell Lines. Metformin Decreases the Expression of NiCl_2_-Upregulated ANGPTL4 in Lung Epithelial Cells and Cancer Cells

We investigated the effects of metformin on nickel-induced oncoprotein expression, given the importance of nickel for tumour progression. An oncology array was used to detect various oncogenic proteins that are induced by nickel exposure and the inhibitory effects of metformin. The array was used to screen the expression levels of 84 cancer-related proteins in NiCl_2_- and metformin-treated BEAS-2B cells, as shown in Figure 3A,B. Of these 84 proteins, 17, including ANGPTL4 and HIF-1α, were significantly upregulated (≥1.25-fold change; Figure 3B) by NiCl_2_ and then immediately reduced by metformin. We further examined the inhibitory effect of metformin on NiCl_2_-treated BEAS-2B and A549 cells. We treated cells with 5 mM metformin on the basis of our previous study, in which metformin was investigated for its chemopreventive effects on the induction of NiCl_2_-induced autophagy. After 24-h treatment, the expression of nickel-induced ANGPTL4 and HIF-1α was significantly suppressed by metformin in both lung epithelial cells and cancer cells (Figure 3C–F). We also verified the expression of the other protein, which was affected by NiCl_2_ and metformin in Oncology Array. As shown in Figure 3C, the expression of carbonic anhydrase IX (CA9) and E-cadherin with coordinate B5, B6 and A19, A20, respectively, exhibited the same trend as Oncology Array. A transwell migration assay was performed to confirm the function of ANGPTL4 and metformin on migration ability in lung cancer cells. The ability of A549 cells to migrate was promoted with increasing doses of recombinant ANGPTL4 treatment (Figure 3G); migration could be blocked by pretreatment with 2.5- and 5-mM metformin for 24 h (Figure 3H). The results suggested that ANGPTL4 reliably promotes lung cancer cell migration and verified the inhibitory effect of metformin.

### 2.3. NiCl_2_ Activates ROS/HIF-1α Generation, and Suppression of HIF-1α Alleviates NiCl_2_-Induced ANGPTL4 Expression

Hypoxia is a characteristic of solid tumours and it is known to promote cancer cell proliferation, invasion, metastasis, angiogenesis, malignancy, and inhibit apoptosis [19]. Recent studies have reported that hypoxia directly upregulates ANGPTL4 and HIF-1α and is one of the main transcriptional regulators of the ANGPTL4 promoter [20,21]. We previously demonstrated that metformin decreases NiCl_2_-activated ROS generation and HIF-1α protein expression [9]. The ROS scavenger N-acetyl-L-cysteine (NAC) was used to inhibit general ROS production to confirm that ANGPTL4 expression is directly regulated by HIF-1α. As apparent in Figure 4A,B, BEAS-2B cells that were pretreated with NAC for 1 h blocked NiCl_2_-induced HIF-1α and ANGPTL4 activation at the highest concentration of 10 mM. We then used the HIF-1α inhibitor PX-478 to suppress HIF-1α translation and transcription under NiCl_2_ exposure; the expression of HIF-1α and ANGPTL4 was effectively decreased (Figure 4C,D). These findings were corroborated while using lentiviral short hairpin RNA (shRNA) to inhibit HIF-1α expression in BEAS-2B cells (Figure 4E,F). Knocking down HIF-1α also largely eliminated the ability of NiCl_2_ to induce ANGPTL4. In addition, the overexpression of HIF-1α without NiCl_2_ treatment still induced ANGPTL4, although the ratio of increase was limited. The blocking effect of metformin was considerable (Figure 4G). These results demonstrate that NiCl_2_-induced HIF-1α protein accumulation is necessary for promoting the expression of ANGPTL4 and that metformin has the ability to inhibit the HIF-1α/ANGPTL4 axis.

### 2.4. NiCl_2_ Promotes ANGPTL4 Expression via the Direct Binding of HIF-1α to the HRE in the ANGPTL4 Promoter Located bp-2.1k and bp-320 from the Transcription Start Site

It has been reported that there are several hypoxia response elements (HREs) in the ANGPTL4 promoter [20,22]. We conducted an ANGPTL4 luciferase activity assay to determine which HRE was the binding site of HIF-1α under NiCl_2_ exposure. The ANGPTL4 luciferase constructs that mutated or deleted the HRE site were all provided by Dr. Inoue and Dr. Wada. Luciferase constructs were transiently transfected into BEAS-2B cells, which were subsequently exposed to NiCl_2_ for 24 h. Among the five constructs, luciferase activity of the HRE deletion and mutation 1 constructs were not significantly amplified under NiCl_2_ exposure (Figure 5A). However, the mutation 2 plasmid was increased, as demonstrated by the significant luminescence intensity, and the mutation 3 plasmid was slightly enhanced. These results demonstrate that the HRE1 region, which is located 2 kb upstream of the transcription start site, is the strongest HIF-1α binding site, because mutations within this site completely disrupted the induction of the ANGPTL4 promoter in the presence of NiCl_2_. We then exposed the HRE1 deletion and WT constructs to NiCl_2_ and PX-478 or NiCl_2_ and metformin for 24 h. Figure 5B illustrates that metformin and PX-478 both inhibited NiCl_2_-induced luciferase activity, but did not affect the activity of constructs that lacked the HRE1 region. After confirming the function of the HRE using the luciferase assay, we conducted a chromatin immunoprecipitation assay to evaluate whether HIF-1α directly binds to HREs and to determine the inhibitory effects of metformin. As shown in Figure 5C, the DNA products were amplified near the HRE1 (−2116 bp to −1931 bp) and HRE3 (−327 bp to −17 bp) regions of the ANGPTL4 promoter. The assay showed that anti-HIF-1α antibody could precipitate HRE1 and HRE3 under NiCl_2_ exposure. The binding ability of HIF-1α at the HRE1 region was stronger than at the HRE3 region, and metformin suppressed the binding of both HREs, as consistent with the luciferase assay.

## 3. Discussion

Most of the studies have reported that ANGPTL4 plays major roles in cancer progression and development and it is involved in the regulation of energy metabolism, angiogenesis, metastasis, and anoikis resistance. However, some studies have indicated that ANGPTL4 has an opposite role in cancer progression. For example, Ushijima et al. demonstrated that ANGPTL4 is a genetically and epigenetically inactivated secreted tumour suppressor, and in gastric cancers ANGPTL4 suppresses in vivo tumorigenesis and angiogenesis through the ERK pathway [23]. It has also been shown that increased ANGPTL4 expression inhibits melanoma and colorectal tumour growth, metastasis, and angiogenesis [24,25]. This contradictory ability of ANGPTL4 in cancer progression might be dependent upon context, tumour type, or tissue/cell-specific post-translational modifications [10].

It has been demonstrated that NiCl_2_ has high invasive potential, which promotes angiogenesis and the acquisition of a persistent mesenchymal phenotype in lung cancer cells, via different pathways [26,27,28]. ANGPTL4 has been recognized as an indicator of whether breast cancer will metastasise to the lung [29], and other studies have indicated that TGF-β primes breast tumours for the seeding of lung metastasis through ANGPTL4 by modulating endothelial integrity to mediate lung metastasis seeding [11]. In our recent investigation, NiCl_2_ increased the level of integrin beta 3 (ITGB3) through TGF-β signalling [27]. In addition, Zhu et al. indicated that tumour-derived ANGPTL4 interacts with integrins to elevate the O2^−^/H_2_O_2_ ratio, thus stimulating tumour development [30]. Although, in this study, we have focused on the tumorigenesis potential that is caused by nickel, due to HIF-1-dependent responses, it seems that TGF-β might participate in NiCl_2_-induced ANGPTL4 performance. As shown in Figure 4E,F, the knockdown of HIF-1α decreased ANGPTL4 expression by half, and decreased it even further when combined with metformin. These findings suggest that ROS are not the only factors that are involved in NiCl_2_-induced ANGPTL4 promotion. We also determined that TGF-β is involved in NiCl_2_-induced ANGPTL4, as shown in Appendix A. SB525334, the TGF-β/Smad inhibitor, decreases NiCl_2_-stimulated ANGPTL4 in both A549 and BEAS-2B cells and metformin decreases it further. This provides evidence that NiCl_2_ induces ANGPTL4 via the co-activation of ROS/HIF-1α and TGF-β, which are two major factors in the central regulation of tumorigenesis.

Several studies indicate that nickel exposure induces oxidative stress through the reduction of antioxidant enzymes and DNA single- and double-strand breaking. The generation of ROS caused by the imbalance of antioxidant or pro-oxidant enzymes leads to the damage of DNA, protein, and lipid, contributes to carcinogenesis [31]. Additionally, mitochondria are considered to be the main sites producing ROS. Mitochondrial dysfunction can interfere with electron transferring and further amplifying more ROS generation [32]. In our previous study, we detected the nickel-induced intracellular ROS production, including nonspecific intracellular esterase, superoxide anions, hydrogen peroxide, or hydroxyl radical while using various ROS-sensitive probes. We also analyzed the expression of oxidative stress-related protein, such as catalase, GPX, NOX, and SOD, and found that nickel decreases the mitochondrial membrane potential [8]. In another study, it has been observed that metformin suppresses nickel-induced ROS generation using H_2_DCFDA staining [9]. This evidence convinces us that the nickel-induced ANGPTL4 or other oxidative stress-respond protein via ROS/HIF-1α are the reason of cell transform into cancer.

In summary, our results define the effect of nickel on ANGPTL4 activation leading to activation through HIF-1α initiation. In addition, we verified the preventive effects of metformin on nickel/hypoxia-induced ROS accumulation in human bronchial epithelial cells. These findings demonstrate the association of ANGPTL4 in nickel-induced lung tumorigenesis and provide a beneficial therapeutic target for slowing the development of lung cancer.

## 4. Materials and Methods

### 4.1. Cell Lines and Chemicals

BEAS-2B cells, the normal human bronchial epithelial cells, were cultured in serum-free LHC-9. MRC-5 cells and WI-38 cells, the normal human lung fibroblasts cell line, were cultured in 10% FBS-MEM. A549, H1299, H1975, CL1-0, and CL1-5 cells, the lung cancer cell lines, were cultured in 10% FBS-DMEM and H1355 were cultured in 10% FBS-RPMI-1640 at 37 °C in an atmosphere of 5% CO_2_. All of the cell lines were purchased from the American Type Culture Collection (ATCC, Manassas, VA, USA) or Bioresource Collection and Research Center (BCRC, Hsinchu, Taiwan). NiCl_2_ (N6136), metformin (D150959), NAC (A9153), and 5-Aza-2’-deoxycytidine (5-aza-dC, A3656) were purchased from Sigma–Aldrich Corporation (St. Louis, MO, USA). PX-478 (10005189) was obtained from Cayman Chemical (Ann Arbor, MI, USA), and recombinant human ANGPTL4 (4487-AN) was purchased from R&D systems (Minneapolis, MN, USA).

### 4.2. Western Blot Analysis

Western blot was performed, as previously described [9]. The primary antibodies used were specific against ANGPTL4 (R&D, AF3485), HIF-1α (BD Biosciences, Franklin Lakes, NJ, USA, 610959), and β-actin (Sigma, A5441, St. Louis, MO, USA).

### 4.3. RNA Isolation, Reverse Transcription-PCR and Quantitative Real-Time PCR

This method was performed, as previously described [9]. Real time-PCR was performed while using ABI StepOnePlus real time PCR system with gene-specific primers and iTaq™ Universal SYBR^®^ Green Supermix (Bio-rad, cat. 172-5121, Hercules, CA, USA). Appendix A shows the primer sets for SYBR Green assays. The quantification of mRNA expression was normalised while using β-actin.

### 4.4. Lentivirus-Based shRNA Silencing

The shRNA lentiviral plasmids and package plasmids were purchased from the National RNAi Core Facility. Individual clones were identified by their unique TRCN (The RNAi Consortium number): shGFP TRCN0000072178 was used as the vector control, shHIF #808 TRCN0000003808 and shHIF #810 TRCN0000195582 targeted HIF-1α. The cells were selected while using 2 μg/mL puromycin for 48 h (Sigma, P8833).

### 4.5. Proteome Profiling

A Proteome Profiler^TM^ Human XL Oncology Array kit (R&D, ARY026) was used to determine the relative levels of 84 human-cancer-related proteins. The cell lysates were harvested using RIPA buffer (2 mM pH 8.0 ETDA, 50 mM Tris pH 7.4, 150 mM NaCl, 1% NP-40, 1 mM PMSF, 1% sodium deoxycholate) containing a protease inhibitor cocktail (Roche, Basel, Switzerland, 04693116001).

### 4.6. Luciferase Reporter Assay

The BEAS-2B cells were transfected in 24-well plates with 0.8 μg of ANGPTL4-promoter-WT-pGL3, ANGPTL4-HRE1-Del, ANGPTL4-HRE1-Mut, ANGPTL4-HRE2-Mut, ANGPTL4-HRE3-Mut, or pGL3-basic vector, together with 0.2 μg of pCMV-β-galactosidase-reporter vector while using Lipofectamine 3000 (Invitrogen, CA, USA, L3000015,). Dr. Tsuyoshi Inoue and Dr. Youichiro Wada kindly provided all of the ANGPTL4 promoter constructs (University of Tokyo) [20]. After 16 h of transfection, the BEAS-2B cells were treated with NiCl_2_, metformin, or PX-478 for 24 h, and then harvested while using Reporter Lysis buffer (Promega, WI, USA). Luciferase activity was normalised with respect to a constitutively expressed β-galactosidase vector. Each transfection was split into several wells to provide triplicates for the luciferase measurement.

### 4.7. Chromatin Immunoprecipitation Assay

The chromatin immunoprecipitation (ChIP) assay procedure was performed according to the manufacturer’s instructions (EZ ChIP™, Chromatin Immunoprecipitation Kit, Merck, 17-371). Briefly, the protein–DNA complexes were cross-linked while using formaldehyde at a final concentration of 1% and then incubated at room temperature for 10 min. Cells were added to 1 mL of 10X glycine to quench the formaldehyde. The cells were washed twice with ice-cold PBS and then collected by centrifugation at 4 °C and resuspended in the cell lysis buffer containing 50 mM Tris–HCl, pH 8, 10 mM EDTA, 1% SDS, and protease inhibitors (PI). Cell lysates were sonicated to shear any cross-linked DNA, generating sizes of approximately 200–1000 bp. The supernatants were diluted with dilution buffer (16.7 mM Tris–HCl, pH 8, 1% Triton X-100, 1.2 mM EDTA, 167 mM NaCl, 0.01% SDS, and PI). Immune complexes were precleared with protein G agarose slurry for 1 h at 4 °C with rotation. After briefly centrifuging at 5000 g at 4 °C for 5 min., 20 μL of the supernatant was removed as the positive control and stored at −20 °C, until used. Subsequently, the remaining supernatant was aliquoted into new microtubes and the immunoprecipitating antibody was added overnight at 4 °C, with rotation. DNA immunoprecipitated with antibodies specific to HIF-1α and mouse IgG was purified and then extracted using the QIAquick PCR purification kit (Qiagen, Hilden, Germany), according to the manufacturer’s instructions. Immunoprecipitated DNA was analysed while using PCR amplification; Appendix A shows the primer.

### 4.8. Bisulfite Treatment and Methylation-Specific PCR (MSP)

Genomic DNA was extracted while using the genomic DNA isolation kit (Qiagen, Hilden, Germany) and treated with sodium bisulfite using an EZ DNA Methylation-Gold kit (Zymo Research, Orange, USA), according to the manufacturer’s recommendation. Bisulfite-converted DNA was amplified using specific ANGPTL4-methylated or ANGPTL4-unmethylated primers (Appendix A). The MSP products were purified and then inserted into the yT&A^TM^ cloning vector (Yeastern Biotech Co. Ltd., Taipei, Taiwan). All of the ligated-DNA was transformed using DH5α competent cells (Yeastern) and sequenced to evaluate the level of methylation at each CpG site.

### 4.9. Statistical Analyses 

Statistical analyses were performed while using SPSS statistical software (version 18.0; SPSS, Inc., Chicago, IL, USA). The data are presented as the mean ± SD of at least three independent experiments, unless otherwise indicated. Statistical significance (*p* < 0.05) was assessed using the two-tailed *t* test or one-way ANOVA. The survival time courses were analysed using the Kaplan–Meier method, and the groups were compared using the log-rank test; *p* < 0.05 was considered to be statistically significant.

## Figures and Tables

**Figure 1 ijms-21-00619-f001:**
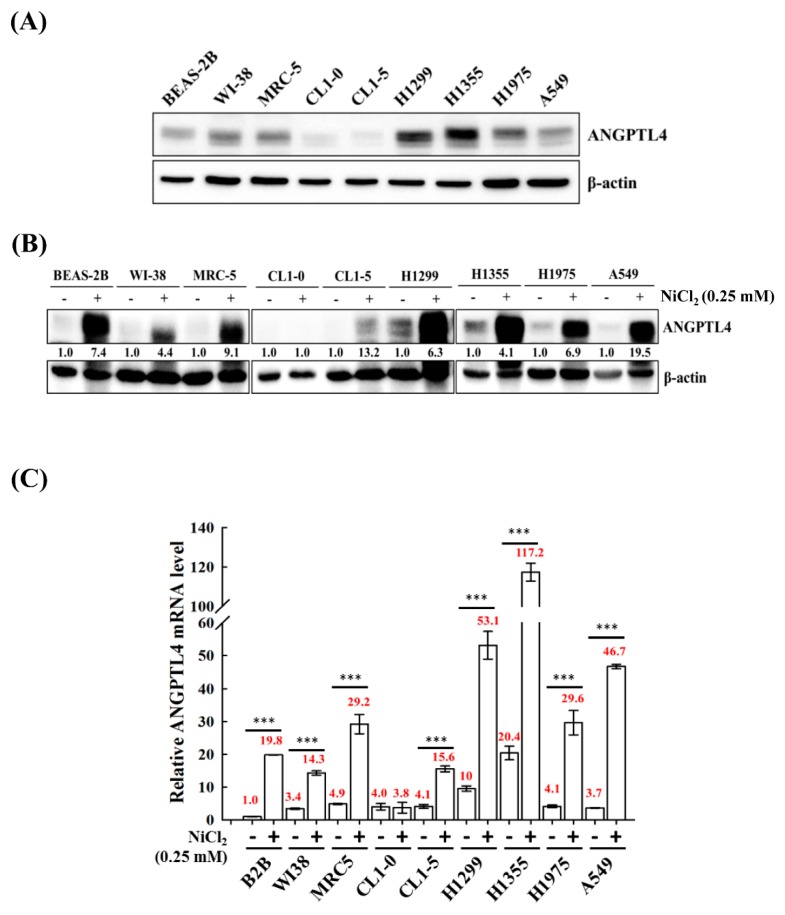
Analysis of ANGPTL4 expression in lung cancer cell lines and normal cell lines exposed to NiCl_2_. (**A**) Protein expression of ANGPTL4 in various normal and tumour lung cells was performed using Western blot analysis. β-actin was used as an internal control. (**B**) ANGPTL4 expression of NiCl_2_ derived from various cancer and normal cell lines by Western blotting and (**C**) real time-PCR. For protein levels, β-actin was used as an internal control. The relative ratio of ANGPTL4 to β-actin is shown. For mRNA levels, quantification of ANGPTL4 was normalised to β-actin, with an average of three independent readings (mean ± SD). *** *p* < 0.001 versus control.

**Figure 2 ijms-21-00619-f002:**
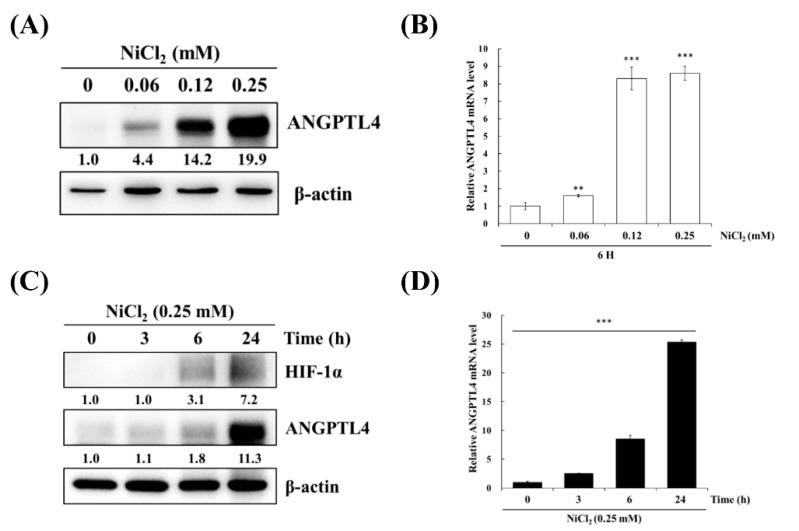
NiCl_2_ activates the expression of ANGPTL4 and HIF1-α in BEAS-2B cells. (**A**) ANGPTL4 protein expression of BEAS-2B cells exposed to NiCl_2_ (0, 0.06, 0.12, and 0.25 mM) for 24 h was performed using western blotting. (**B**) The mRNA level of cells exposed to NiCl_2_ for 6 h was performed by real time-PCR. Significant differences from the untreated cells are indicated by ** *p* < 0.01 or *** *p* < 0.001 (**C**,**D**) Time-course analysis was also performed on BEAS-2B cells, which were incubated with 0.25 mM NiCl_2_ for 0, 3, 6, and 24 h.

**Figure 3 ijms-21-00619-f003:**
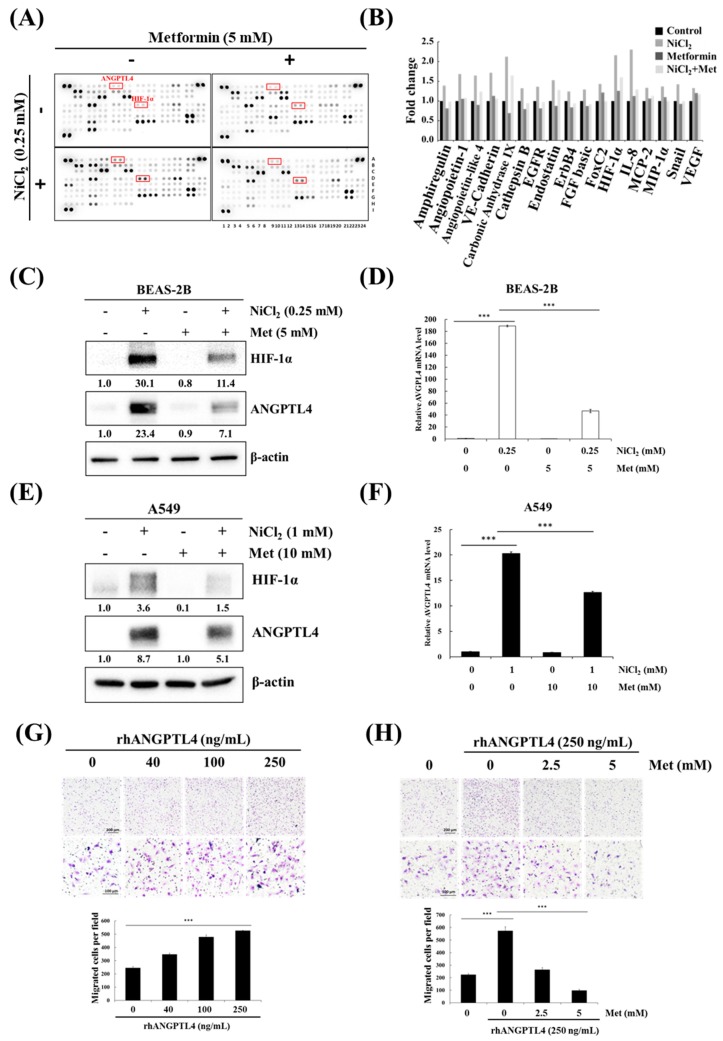
Metformin represses NiCl_2_-induced ANGPTL4 activation and hypoxia-inducible factor-1α (HIF-1α) expression. (**A**) Protein profiles were performed in NiCl_2_- and metformin-treated BEAS-2B cells for 48 h using a Proteome Profiler Human XL Oncology Array kit. (**B**) Seventeen gene expression values were generated by calculating the mean spot pixel density from the array that was affecting by NiCl_2_ and metformin. Data are presented as a fold change in each protein compared with the untreated group. (**C**,**D**) BEAS-2B and (**E**,**F**) A549 cells combined with 0.25 mM or 1 mM NiCl_2_ and 5 mM or 10 mM metformin for 24 h, western blotting and qPCR were used to determine protein and mRNA expression. (**G**) Migration of A549 cells across transwell filters for 24 h. A quantity of 10% FBS DMEM medium was placed in the bottom chamber containing various doses of ANGPTL4 recombinant protein. Bar graph: Quantitation of migrated cells/field. The results were obtained from three random fields per filter and represented the means ± SD. *** *p* < 0.001 (**H**) Cells were pretreated with 0, 2.5, and 5 mM metformin for 24 h and then seeded into transwell chambers without metformin; the bottom chamber contained 25 ng/mL ANGPTL4 recombinant protein.

**Figure 4 ijms-21-00619-f004:**
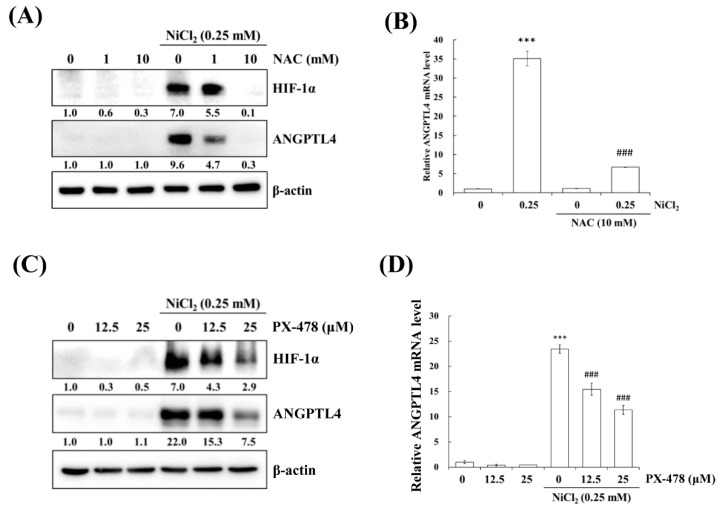
HIF-1α is involved in NiCl_2_-induced ANGPTL4 expression. (**A**) N-acetyl-L-cysteine (NAC), the ROS scavenger, inhibited NiCl_2_-upregulated ANGPTL4 expression. Determination of HIF-1α and ANGPTL4 protein levels in BEAS-2B cells pretreated with NAC for 1 h and cultured under NiCl_2_ for 24 h. (**B**) qPCR was performed to detect ANGPTL4 mRNA levels. Data are shown relative to untreated cells after normalisation to β-actin and reported as the mean ± SD of triplicate experiments. *** *p* < 0.001 compared with untreated cells; ^###^
*p* < 0.001 compared with NiCl_2_-treated cells, as determined by two-tailed *t* tests. (**C**) Cells were pretreated with PTX-478, the HIF-1α transcriptional inhibitor, for 1 h and then exposed to NiCl_2_ for 24 h. Western blotting and (**D**) qPCR were performed to determine the protein and mRNA expression. (**E**) BEAS-2B cells were treated with NiCl_2_ and metformin for 48 h after infecting with lentivirus carrying shGFP (vector control) or shHIF-1α (clone no. #808 and #810). The protein levels of HIF-1α, ANGPTL4, and β-actin were determined while using Western blot analysis. β-actin was used as the internal control. (**F**) Quantitative analysis of ANGPTL4 mRNA was performed using qPCR. The data are presented as the mean ± SD of triplicate experiments; a = *p* < 0.001 compared with shGFP untreated cells; b = *p* < 0.001 as compared with shGFP Ni-treated cells; c = *p* < 0.05 compared with shGFP Ni- and metformin cotreated cells. (**G**) HIF-1α overexpression promoted ANGPTL4, which was then inhibited by metformin. Western blot analysis of BEAS-2B cells overexpressing pcDNA3 or HA-HIF-1α-pcDNA3.

**Figure 5 ijms-21-00619-f005:**
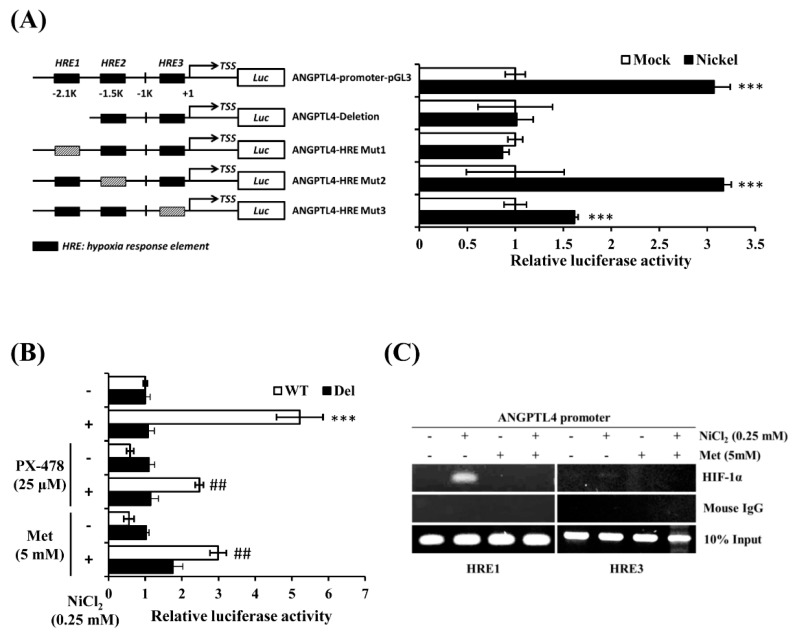
Metformin represses NiCl_2_-induced ANGPTL4 by obstructing the binding of HIF-1α to its promoter. (**A**) The luciferase reporter assays were used to analyse the effects of deletion or mutation of the hypoxia response element (HRE) on the ANGPTL4 promoter activity of cell extracts. BEAS-2B cells were transfected with wild-type, HRE deletion, or mutant ANGPTL4 promoter constructs and cultured under NiCl_2_ for 24 h. Luciferase activities were normalised to the respective β-gal activities. Data are presented as the mean ± SD of three independent experiments performed in triplicate; *** *p* < 0.001 compared with each construct control. (**B**) The wild type or deletion luciferase plasmids were co-transfected with β-gal into BEAS-2B cells. After 16 h of NiCl_2_ incubation with PTX-478 or metformin, luciferase activity/β-gal activity was measured. Data are presented as the mean ± SD; *** *p* < 0.001 compared with the untreated WT group; ^##^
*p* < 0.01 compared with the untreated Del group. (**C**) Chromatin immunoprecipitation assays were performed to determine binding of HIF-1α to the HRE1 and HRE3 sites located in the ANGPTL4 promoters. The anti-HIF-1α antibody and mouse IgG were used for immunoprecipitation with DNA isolated from BEAS-2B cells that were treated with NiCl_2_ and metformin for 24 h. The immunoprecipitation was amplified while using PCR with primers that covered the HRE. WT, wild-type ANGPTL4 promoter construct. Del, HRE deletion ANGPTL4 promoter construct.

## Data Availability

The data sets used and/or analyzed during the current study are available from the corresponding author on reasonable request.

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
