# Peer review of "Metformin Mitigates Nickel-Elicited Angiopoietin-Like Protein 4 Expression via HIF-1α for Lung Tumorigenesis"

_ijms, 2020, doi:10.3390/ijms21020619_

Round 1

Reviewer 1 Report

The present study evaluated the effect of nickel on ANGPTL4 activation leading to activation through HIF-1α initiation and verified the preventive effects of metformin on nickel/hypoxia-induced ROS accumulation in human bronchial epithelial cells.

The study design and results are well-described and their finding have a clinical implication not only in the field of oncology, but also metabolism.

I have only minor comments to the paper:

In the methods and data presentation, normal cell lines may be indicated or listed separately.

And, results also need to be described with a brief description; not all readers are oncologist or pulmonologist. e.g. BEAS-2B, a normal cell line of human bronchial epithelium.

Lines 36-38 may be referenced.

There is a typo in line 299

Reviewer 2 Report

The manuscript submitted by Kang et al demonstrated that the axis of Nickel/HIF-1α/ANGPTL4 in lung cells. This novel pathway was thought to be one of the potential of explanations of nickel exposure contributes to lung cancer progression. In addition, they also verified the potential preventive effects of metformin on this pathway. The study shed some important insights into carcinogen effect of Nickel and potential of therapeutic effect of metformin. Before this manuscript can be considered for publication, many concerns should be addressed:

Limitations:

1.The present study is only focusing on the molecular mechanism(s), authors should provide phenotype of these molecular events. At least, the evidence of the effect of this pathway on the transition of normal lung cells to lung cancer should be provided.

2.In Figure 3A, authors utilized Proteome Profiler Human XL Oncology Array kit to screen altered protein. Among them, ANGPTL4 has already been verified by western blot. Authors should re-confirm other changed protein by western blot.

3.To generalize the conclusion of the present study, it is better to stimulate HIF-1α expression other than nickel exposure. By doing this, the significance of HIF-1α/ANGPTL4 pathway can be generalized into different physiopathological context.

4.The potential mechanism of nickel on HIF-1α/ROS accumulation should be thoroughly discussed or investigated.

Round 2

Reviewer 2 Report

The revised manuscript meets the criteria of IJMS. I agree the current manuscript could be accepted for publication.